# LoLoRA: Locally Fine-Tuned Low Rank Adapters

## Abstract

Low-Rank Adaptation (LoRA) is a memory-efficient fine-tuning method for large language models (LLMs) that approximates weight updates as $\Delta W = BA$, where $B \in \mathbb{R}^{n \times r}$, $A \in \mathbb{R}^{r \times m}$ and $r \ll \min(m, n)$. To maximize memory savings, one can freeze matrix A, avoiding the storage of its input activations, but this often degrades performance. In this work, we mitigate this trade-off by introducing gradient-free updates to matrix A during the forward pass. Our method computes these updates based on the layer's immediate input, allowing it to adapt to input distribution shifts without storing activations for the backward pass. This approach maintains performance comparable to standard LoRA while further reducing the memory required for fine-tuning.

## 1 Introduction

Transformer-based LLMs are increasingly applied across different domains, going beyond simple language understanding problems. While LLMs can solve a wide range of reasoning problems given sufficient context, specialized fine-tuning typically further improves their performance on downstream tasks. In typical training settings, full fine-tuning is comparable to pre-training in terms of memory requirements, as both require storing activations and optimizer states, even though the fine-tuning dataset is much smaller. At the same time, the compute and resource costs of training and fine-tuning models have increased significantly in recent years. Therefore, developing more efficient training methods is critical to reduce time and computational resources, especially memory consumption.

There is a group of methods called parameter-efficient fine-tuning (PEFT), which focuses on modifying a smaller subset of parameters, enabling effective adaptation to new tasks without the computational overhead of tuning the entire model (Han et al., 2024). Among PEFT methods, LoRA (Hu et al., 2021) stands out as one of the most widely adopted and extensively studied approaches. Low-Rank Adaptation (LoRA) is a memory-efficient fine-tuning method for large language models (LLMs) that approximates weight updates as $\Delta W = BA$, where $B \in \mathbb{R}^{n \times r}$, $A \in \mathbb{R}^{r \times m}$ and $r \ll \min(m, n)$, $A$ is initialized with uniform noise, and $B$ with zeros. To further develop the LoRA method, we propose a modification that allows us to eliminate the need to store activations in the adapter and thus reduce the memory requirements for fine-tuning. A naive implementation of this idea is LoRA-FA (Zhang et al., 2023b), where matrix $A$ is frozen during training, which saves memory on activations and optimizer state. In our work, in contrast, we train adapter $A$ on the forward pass, relying only on layer inputs, which also eliminates the need to store activations for backpropagation. Further, we refer to this kind of forward-pass updates, which don't require backpropagation, as local learning rules. We compare different initializations of $A$ in freezing mode (LoRA-FA) and explore several variants of local learning rules of adapter $A$.

Also, we derive a theoretical insight that is in line with a data-driven initialization method (EVA) proposed by Paischer et al. (2024). In that work, it was experimentally shown that initializing $A$ with principal component vectors of the pretrained model's activations on a downstream task outperforms other known initialization techniques. We theoretically prove, however, for a wider class of matrices, that it is the optimal initialization under certain assumptions. The key difference of the proposed method (*LoLoRA*) from EVA is that it is proposed to use local learning rules to iteratively adapt $A$ during fine-tuning to the optimal subspace.

The structure of the paper is as follows. Section 2 provides a brief overview of existing modifications of LoRA methods, and reviews the current state of research on localized learning relevant to this paper. Sections 3 and 4 contain a description of the proposed method and its theoretical justification. Section 5 introduces experimental results of the proposed method. Main takeaways and directions for further research are drawn in the Conclusion.

## 2 RELATED WORK

Among the parameter-efficient fine-tuning (PEFT) methods, LoRA has gained the greatest practical popularity due to its simplicity of integration and efficiency. Many modifications based on it have appeared, aimed either at quality improvement or at more efficient memory utilization.

A line of work (Meng et al., 2024; Büyükakyüz, 2024; Wang et al., 2025) focuses on the "informed" initialization of the adapter matrix $A$ from the properties of the pre-trained matrix $W$, to which the update $\Delta W = BA$ is applied. The general hypothesis is that the internal structure of $W$ correlates with the direction of its change when adapting to the downstream task. The alternative branch (Zhao et al., 2024; Wang et al., 2024) initializes $A$ via the SVD of the gradient $\nabla W$, assuming that the principal components of the gradient define a successful starting subspace for the downstream optimization of $\Delta W = BA$. In Paischer et al. (2024), the adapter is built based on PCA of the layer input representations; in addition, a fraction of the explained variance is used to dynamically distribute ranks across layers, allowing additional memory to be spent more efficiently.

The idea of adaptive ranking is also developed in Zhang et al. (2023a;c); Renduchintala et al. (2024), where the rank value is selected or changed during training depending on the importance of the layer or the training signal. In parallel, approaches to memory saving by sharing parameters between adapters and/or layers are being investigated - see Kopiczko et al. (2024); Renduchintala et al. (2024); Song et al. (2025).

Another area of fine-tuning cost reduction is to shift some of the computation to local updates that don't require costly backpropagation through whole model, which has not yet been much explored in the context of LLM. However, a step towards this direction was made in (Key et al., 2023). The authors proposed to divide the Transformer into chunks of several layers, which are all trained to predict the next output token based on their local input. It turns out that fine-tuning of two halves of the model separately is better than fine-tuning only the last half of the layers while keeping the first half frozen. Along these lines, several works (Nøkland & Eidnes, 2019; Wang et al., 2021; Apolinario et al., 2024) demonstrate the use of local per-layer losses to train Convolutional Neural Network (CNN) classifiers with SGD. It forms a foundation for the local learning method based on the Transformer's skip connection use–another promising direction for efficient fine-tuning.

Other works on local learning are devoted to CNN layers. Studies such as Lagani et al. (2022) suggest that Hebbian and SGD-based learning optimize interfering objectives. When some layers are trained using SGD and others with Hebbian-like updates, classification accuracy drops significantly – except when local rules are applied exclusively to the outermost layers. Similarly, Krithivasan et al. (2022) shows that switching between SGD and weakly supervised Hebbian updates in a layer-by-layer manner does not yield significant improvements compared to simply freezing weights. That is, it remains a challenge to reconcile local learning and end-to-end backpropagation in one method, which we also address in our work.

## 3 METHOD

In this section, we present the proposed method. We begin with preliminaries and a short recap of LoRA. In Section 3.2, we introduce our variant, *LoLoRA*, designed to reduce activation memory by combining local unsupervised updates of adapter $A$ with gradient-based updates of adapter $B$. Finally, in Section 3.3, we describe the training procedure and provide the algorithmic formulation of our method.

### 3.1 PRELIMINARIES AND LoRA

**Definition 3.1 (Submodule)** *The model submodule is a linear mapping of the form $h = Wz$, where $z$ is the submodule's local input, and $W$ can be either $W_q$, $W_k$, $W_v$, or $W_o$ in an attention layer, or $W_{proj}^{(up)}$, $W_{proj}^{(down)}$ in an MLP layer.*

**Revisiting LoRA.** LoRA (Hu et al., 2021) is based on the idea that the weight change matrix is usually low-rank $\Delta W = W' - W$, where $W$, $W'$ are the pretrained and fine-tuned Transformer-based LLM weight matrices. Therefore, it was proposed to approximate $\Delta W = BA$ for each submodule, where $B \in \mathbb{R}^{n \times r}$, $A \in \mathbb{R}^{r \times n}$ are the adaptation matrices with rank $r \ll n$. That is, LoRA allows training only $2rn$ parameters for each submodule, instead of $n^2$ (for notational convenience, throughout this paper we assume all matrices are square).

**Why LoRA is attractive.** The standard LoRA fine-tuning method has demonstrated strong performance in various tasks, excelling in terms of final model quality, efficiency, and memory usage during training. The LoRA method significantly reduces additional parameters required for fine-tuning, thus minimizing memory consumption associated with storing additional weights and optimizer statistics.

**Bottleneck: retained activations.** However, vanilla LoRA does not reduce activation memory needed for backpropagation through the LoRA-augmented linear layers. This issue is addressed by the LoRA-FA method, which suggests randomly initializing adapter matrix $A$ and freezing it during training. This approach significantly cuts memory costs for storing hidden states. Nevertheless, as will be demonstrated experimentally, the LoRA-FA method has limitations in performance due to the suboptimal feature extraction by a randomly initialized low-rank matrix $A$, especially if hidden states exhibit inherently low effective dimensionality – a common scenario in large Transformer models (Valeriani et al., 2023).

### 3.2 OUR APPROACH: *LoLoRA*

We propose a hybrid approach, *LoLoRA*, dynamically adapting to the input distribution through local updates of LoRA matrix $A$ (see Figure 1), also significantly reducing the memory needed for input storage. Our method aims to increase LoRA fine-tuning efficiency by freeing $A$ from costly gradient-based updates. It was shown that freezing $A$ matrix during fine-tuning does not influence much the overall LoRA's performance (Zhang et al., 2023b). In (Zhu et al., 2024), it was also shown that matrix $B$ is the most responsible for adapting to a downstream task and $A$ matrices are more similar between different tasks. We make the next step further by showing that just random initialization might not be the best for $A$ (see Section 4). Specifically, we derive (see Theorem 4.4) a set of optimal $A$ initializations, which is a set of arbitrary nonsingular linear transformations of $r$ principal components of the submodule's input covariance matrix. Following this theoretical result, we propose a hybrid method for LLM LoRA fine-tuning, which combines local unsupervised updates of the $A \in \mathbb{R}^{r \times n}$ matrix and gradient descent updates of $B \in \mathbb{R}^{n \times r}$ with backpropagation through the model's output loss.

### 3.3 TRAINING PROCEDURE

For each Transformer's submodule, we train LoRA adapters as shown in Algorithm 1. Matrix $A$ is updated during the forward pass based on Hebbian Principal Component Analysis (HPCA), specifically using the Subspace Network learning (SNL) algorithm (Oja, 1989) or by minimizing local symmetric autoencoder (AE) loss (see Definition 4.3). HPCA methods have been shown to converge to principal components of input data or eigenvectors corresponding to the largest eigenvalues of the input covariance matrix (Oja, 1992), and SNL was selected for its simplicity and favorable convergence properties (see also Appendix B for its convergence analysis).

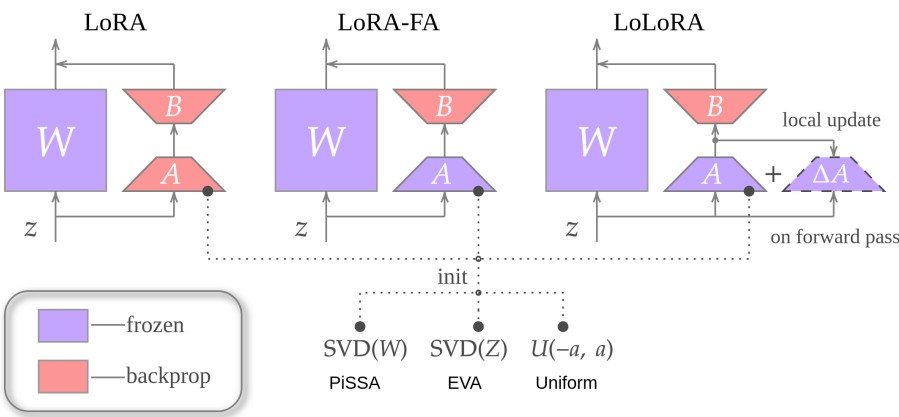

Figure 1: Diagram of our memory efficient method *LoLoRA* and considered baselines.

---

**Algorithm 1** *LoLoRA*

---

**Input:** input $z$; frozen base $W$; adapters $A, B$; local rule LocalRule; optimizer $\text{Opt}_{\text{loc}}$ (for $A$)
**Output:** output $h$

1: $u \leftarrow Az$
2: $g_A^{\text{loc}} \leftarrow \text{LocalRule}(A, z, u)$
3: $\text{Opt}_{\text{loc}}.\text{accumulate}(A, g_A^{\text{loc}})$ # update local optimizer state
4: $\text{Opt}_{\text{loc}}.\text{step}(A)$ # immediate step on $A$
5: $h \leftarrow Wz + Bu$
6: FREE_MEMORY$(z)$ # do not retain $z$ for $A$'s backward; keep $u$ for $B$
7: **return** $h$

---

## 4 RATIONALE

In this section, we develop the theoretical motivation for our method. We begin by formalizing fine-tuning as the minimization of global language modeling loss. Then we reformulate the problem of LoRA tuning as a low-rank regression, and analyze optimal initializations of LoRA adapters under random regression assumptions. All formal proofs are deferred to Appendix A.

**Preliminary definitions.** The goal of fine-tuning is to minimize conditional language modeling loss (global loss) on a downstream task Vaswani et al. (2023).

**Definition 4.1 (Global Loss)** *Let* $P_\Phi(y \mid x)$ *denote a Transformer-based language model, parametrized by* $\Phi$. *Global loss is calculated for the model's outputs as follows:*

$$\mathcal{G}(\Phi, D) = -\mathbb{E}_{(x,y) \sim p_D} \sum_{t=1}^{L} \log(P_\Phi(y_t \mid x_{\leq t})) \tag{1}$$

*where* $p_D$ *is the data distribution for task* $D$, $L$*–context length,* $x_t, y_t$*–input and output tokens.*

Let's reformulate this problem as a linear regression through the submodule's local target.

**Definition 4.2 (Submodule's Local Target)** *Let us denote* $z$*–submodule's local input, and* $h$*– submodule's output. Then for an optimally fine-tuned model that minimizes the global loss, the output of each submodule can be written as* $h = W_0 z + \Delta W_0 z$, $W_0$*–pretrained weights,* $\Delta W_0$ *– submodule's optimal weight change. We call* $\tau = \Delta W_0 z$ *the submodule's local target for optimal fine-tuning for a specific downstream task.*

That is, if the LoRA fine-tuned model minimizes global loss for a downstream task, the LoRA adapters of each submodule approximate $\Delta W_0$.

**Low-rank Regression Formulation.** Consider the problem of low-rank linear regression

$$L(A, B) = \mathbb{E} \frac{1}{2} ||\tau - BAz||^2, \tag{2}$$

where $z, \tau \in \mathbb{R}^n$ – input and target vectors, respectively, $A \in \mathbb{R}^{r \times n}, B \in \mathbb{R}^{n \times r}$ – low-rank matrices.

Let us introduce some notation: $W = BA$ is a linear regression matrix, $\Sigma_{zz} = \mathbb{E}zz^\top$ is an input covariance matrix. $\Sigma_{\tau z} = \mathbb{E}\tau z^\top$ – target-input correlation matrix, and $\Sigma_{z\tau} = \Sigma_{\tau z}^\top$ – input-target correlation matrix.

We are interested in solving problem 2 when $A$ is fixed, referring to LoRA with a frozen $A$ adapter. Let us consider how to properly initialize the adapter $A$ under the assumption that the target is unknown. To this end, we assume a random linear regression problem of the form $\tau = \Delta W_0 z$, where $[\Delta W_0]_{ij} \sim \mathcal{N}(0, \sigma^2)$. Let $B = B(A)$ be the *optimal* solution to problem 2 with respect to $B$, given fixed $\Delta W_0$ and $A$. We call $A$ *optimal* if it minimizes the expected loss function $g(A) = \mathbb{E}_{\Delta W_0} L(A, B(A))$.

**Assumption 4.1 (Random regression matrix)** *Let* $\Delta W_0 \in \mathbb{R}^{n \times n}$. *Its entries are i.i.d. Gaussian:* $(\Delta W_0)_{ij} \overset{i.i.d.}{\sim} \mathcal{N}(0, \sigma^2)$ *for some* $\sigma^2 > 0$.

**Assumption 4.2 (Input correlation matrix decomposition)** *Suppose that*

$$\Sigma_{zz} = Q \operatorname{diag}(\lambda_1, \ldots, \lambda_n) Q^\top,$$

*where* $\lambda_1 \geq \ldots \geq \lambda_n$ *and* $Q^\top Q = I_n$. *In these notations* $\lambda_i$ – *ith eigenvalue, and the columns of matrix* $Q$ *are the corresponding eigenvectors.*

**Remark 4.3 (Validity of notations)** *Despite the ambiguity of* $B(A)$ *when* $A$ *is not fully ranked, it should be noted that the notion of* $\mathbb{E}_{\Delta W_0} L(A, B(A))$ *is valid due to its measurability:*

$$g(A) = \mathbb{E}_{\Delta W_0} \inf_B \mathbb{E}_z \frac{1}{2} ||\Delta W_0 z - BAz||^2.$$

Here $\inf_B \mathbb{E}_z \frac{1}{2} ||\Delta W_0 z - BAz||^2$ denotes the minimum of $L(A, B(A))$ with respect to $B$, given $A$ is fixed. Expectation over $\Delta W_0$ tells that $g(A)$ doesn't include information about targets.

**Theorem 4.4 (Exact description of the set of optimal** $A$**)** *Assume that Assumptions 4.1 and 4.2 hold. Then the following statements are true:*

*(i)* $\min_A g(A) = \frac{1}{2}\sigma^2 n \sum_{k=r+1}^n \lambda_k$.

*(ii)* *Let* $d = \min\{r, \operatorname{rk}(\Sigma_{zz})\}, \mathcal{O}(n, d) = \{V \in \mathbb{R}^{n \times d} \mid V^\top V = I_d\}, \mathcal{V} = \arg\max_{V \in \mathcal{O}(n,d)} \operatorname{tr}(V^\top \Sigma_{zz} V)$, *then the following holds:* $\arg\min_A g(A) = \{A \mid \exists C \in \mathbb{R}^{r \times d}, V \in \mathcal{V} : A\Sigma_{zz}^{1/2} = CV^\top, \operatorname{rk}(C) = d\}$.

*(iii)* *In the case* $\lambda_1 > \ldots > \lambda_n > 0$, *the following holds:* $\arg\min_A g(A) = \{CQ_{*,:r}^\top \mid C \text{ is nonsingular}\}$.

Theorem 4.4 shows that if information about targets is not available, an optimal $A$ is an arbitrary nonsingular linear transformation of the covariance matrix's first $r$ eigenvectors. This is exactly the form of $A$ that HPCA updates converge to according to (Oja, 1992). Moreover, SGD or Adam updates through the model's global loss ensure optimal $B$.

Now consider a slightly different problem: let $B$ be initialized and frozen, and let $A$ be optimized by gradient descent. Let $A = A(B)$ be the *optimal* solution to problem 2 with respect to $A$, given fixed $\Delta W_0$ and $B$. We define $B$ as *optimal* if it minimizes the expected loss function $h(B) = \mathbb{E}_{\Delta W_0} L(A(B), B)$.

**Theorem 4.5 (Exact description of the set of optimal $B$)** *Assume that Assumptions 4.1 and 4.2 hold. Then the following statements are true:*

*(i)* $h(B) = \frac{1}{2}\sigma^2(n - \text{rk}(B))\sum_{k=1}^{n}\lambda_k.$

*(ii)* $\arg\min_B h(B) = \{B \mid \text{rk}(B) = r\}.$

Theorem 4.5 shows that any full-rank initialization of matrix $B$ gives the same result.

We can also see that if the input correlation matrix is very unbalanced (the first few eigenvalues are much larger than the rest of the spectrum), then the proper initialization of $A$ provides stronger advantage than at a balanced spectrum.

**Implications.** Based on these results, we can conclude that under certain conditions on the problem, initialization via PCA decomposition of input vectors is optimal. Also, these results highlight the asymmetry of adapters $A$ and $B$ under known information about the input distribution - there is a "good" initialization for adapter $A$, but none for adapter $B$.

This theoretical insight complements the results of EVA paper (Paischer et al., 2024) as well as the paper (Zhu et al., 2024) on the asymmetry of adapters $A$ and $B$, where it is shown that random initialization of adapter $A$ followed by freezing is, with high probability, better than random initialization of adapter B followed by freezing.

### 4.1 LoLoRA AUTOENCODER

According to Theorem 4.4, adapter $A$ can be trained by any of the methods of maximum eigenspace approximation, not only HPCA.

Let's consider another option of learning the adapter $A$ locally.

**Definition 4.3 (LoLoRA Autoencoder)** *We call hybrid LoRA AE a method in which the matrix $A$ is updated by local gradient, minimizing $\mathbb{E}_z \frac{1}{2}||z - A^\top A z||^2$, while $B$ is updated to approximate the submodule's local target $\mathbb{E}_{(z,\tau)}\frac{1}{2}||\tau - BAz||^2$.*

Considering *LoLoRA AE*, it can be shown that minimizing $\mathbb{E}_z \frac{1}{2}||z - A^\top A z||^2$ converges to a matrix $A$ whose row space spans the dominant eigensubspace of the covariance matrix $\Sigma_{zz}$.

**Theorem 4.6** *Assume $\Sigma_{zz}$ is non-singular. Let $l(A) = \mathbb{E}_z||z - A^\top A z||^2$. Then the following statements are satisfied:*

*(i) Any local minimum of $l(A)$ is global.*

*(ii)* $\arg\min_A l(A) = \{V^\top \mid V \in \arg\max_{V \in \mathcal{O}(n,r)} \text{tr}(V^\top \Sigma_{zz} V)\}\}.$

See proof in Appendix A

Table 6 will compare HPCA and AE. This data shows that HPCA is slightly superior to AE.

## 5 EXPERIMENTS

In this section, we evaluate the proposed method *LoLoRA* on several scenarios and perform ablations. First, we examine text understanding tasks on a GLUE (Wang et al., 2019) subset with the RoBERTa-large[1](Liu et al., 2019) model. Then, we test its reasoning performance by fine-tuning LLaMA-3.1-8B-Instruct[2] on MetaMathQA (Yu et al., 2023) with GSM8K Platinum (Cobbe et al., 2021; Vendrow et al., 2025) tests. Next, we consider multimodal fine-tuning LLaVA-v1.5-7B (Liu et al., 2023b;a; 2024b), training one epoch on 20% subset (30k train + 1.5k validation) LLaVA Visual Instruct 150K (Liu et al., 2024a). Finally, we perform ablations on TinyLlama-1.1B[3](Zhang et al., 2024) with Alpaca dataset (Taori et al., 2023).

---

[1]huggingface.co/FacebookAI/roberta-large

[2]huggingface.co/meta-llama/Llama-3.1-8B-Instruct

[3]huggingface.co/TinyLLaMA/TinyLLaMA-1.1B-Chat-v1.0

**Tasks and datasets:**

- GLUE (CoLA, RTE, MRPC, STS-B, MNLI, QNLI, QQP, SST-2). Classic NLU tasks; reporting metrics: accuracy/Matthews/Pearson.
- MetaMathQA → GSM8K Platinum. Fine-tuning to predict correct solution text and test matching answers.
- LLaVA Visual Instruct 150K (20%). "Question-image → text" instructions; validation on the deferred portion of the same instruction/image pool.
- Alpaca (for ablations). A small instruction set; used as a fast testing ground for comparing update and initialization variants.

**Baselines and Methods**

- LoRA. Gradient learning of matrices $A, B$ via backpropagation.
- LoRA-FA (Frozen $A$). Matrix $A$ is frozen; only $B$ is trained.
- *LoLoRA* (ours). $A$ is locally updated in the forward pass without storing inputs; $B$ is updated using a regular backprop. Where appropriate, we additionally show different initialization variants (e.g., EVA) and local rules.

**General Training Settings**

Same for all scenarios : AdamW ($\beta_1 = 0.9$, $\beta_2 = 0.999$), dropout=0.0, weight_decay=0.0. Using warmup. We train all attention linear layers $W_q, W_k, W_v, W_o$. Hardware: one NVIDIA H100. For more details on hyperparameters for each experiment, see Appendix C.

## 5.1 Natural Language Understanding: RoBERTa-large on GLUE

To evaluate the performance of our approach on text-only understanding tasks, we fine-tune RoBERTa-large on several GLUE benchmarks: CoLA, RTE, MRPC, STS-B, MNLI, QNLI, QQP, and SST-2. We compare classical LoRA, LoRA-FA with uniform and EVA initialization, and our *LoLoRA* method with HPCA updates. Results are reported in Tables 1 and 2. We show Matthews correlation for CoLA, Pearson correlation for STS-B, matched accuracy for MNLI and accuracy for others. We also analyze methods' memory efficiency in Appendix D.

Table 1: Performance of LoRA variants on RoBERTa-large across GLUE tasks (part 1).

| Method | CoLA | RTE | MRPC | STS-B |
|---|---|---|---|---|
| LoRA (uniform) | **69.6±0.5** | 84.7±2.1 | **90.9±0.4** | **92.3±0.1** |
| LoRA-FA (uniform) | 67.9±0.9 | **86.4±1.1** | 89.8±0.8 | 92.0±0.3 |
| LoRA-FA (EVA) | 64.7±0.6 | 83.6±2.3 | 90.0±0.5 | 91.9±0.1 |
| *LoLoRA HPCA* | 66.3±1.3 | 84.6±1.6 | 89.9±0.4 | 92.0±0.3 |

Table 2: Performance of LoRA variants on RoBERTa-large across GLUE tasks (part 2).

| Method | MNLI | QNLI | QQP | SST-2 |
|---|---|---|---|---|
| LoRA (uniform) | **90.8±0.1** | **94.9±0.1** | **91.7±0.0** | 96.6±0.1 |
| LoRA-FA (uniform) | 90.6±0.1 | 94.6±0.2 | 90.8±0.1 | **96.7±0.2** |
| LoRA-FA (EVA) | 90.4±0.1 | 94.5±0.0 | 90.6±0.1 | 96.3±0.1 |
| *LoLoRA HPCA* | 90.3±0.1 | 94.7±0.0 | 90.6±0.0 | 96.4±0.2 |

**Summary.** On GLUE, classical LoRA remains the strongest overall. Freezing $A$ (LoRA-FA) can sometimes improve stability (RTE, SST-2), but often reduces performance. EVA (Explained Variance Adaptation from (Paischer et al., 2024)) initialization underperforms on this setting, while *LoLoRA* achieves slightly better results than LoRA-FA (EVA). Both LoRA-FA and *LoLoRA* show up to 20% less GPU memory requirements in comparison to standard LoRA (see Appendix D).

## 5.2 MATHEMATICAL REASONING: LLaMA-3.1-8B ON METAMATHQA

This section details the fine-tuning results of LLaMA-3.1-8B-Instruct on the MetaMathQA dataset, with evaluation performed on the GSM8K Platinum benchmark. As in the previous section, we compare classical LoRA, LoRA-FA with uniform and EVA initialization, and *LoLoRA HPCA*. The results of one-epoch fine-tuning are reported in Table 3, presenting GSM8K test accuracy averaged over three fine-tuned models with different seed values. The model was tested on GSM8K every 0.2 epoch during fine-tuning, and the best result is reported for each method.

Table 3: LLaMA-3.1-8B-Instruct (MathQA) Fine-Tuning Results on GSM8K Platinum Benchmark

| Method | Accuracy | Extra Memory (GB) |
|---|---|---|
| LoRA (unifrom) | 0.821 ±0.005 | 30 |
| LoRA-FA (unifrom) | 0.826 ±0.005 | **26** |
| LoRA-FA (EVA) | **0.829 ±0.005** | **26** |
| *LoLoRA HPCA* | **0.829 ±0.004** | **26** |
| Base Model | 0.79 | - |

**Summary.** As can be seen from Table 3, LoRA-FA and *LoLoRA HPCA* achieve approximately 13% extra memory reduction (26 GB vs 30 GB, peak allocated memory, excluding model size) compared to standard LoRA while maintaining or improving performance. Both LoRA-FA with EVA initialization and *LoLoRA HPCA* achieve the highest accuracy of 82.9%, representing a 3.9 percentage point improvement over the base model.

## 5.3 MULTIMODAL FINE-TUNING: LLaVA-v1.5-7B (VISUAL INSTRUCT 150K)

To further evaluate the versatility of our method, we fine-tuned the multimodal model LLaVA-v1.5-7B. The model was trained for one epoch on a 20% subset (30k samples) of the LLaVA Visual Instruct 150K dataset, with 1.5k samples held for validation.

We compare several baselines: standard LoRA with uniform initialization, LoRA with EVA initialization, their frozen-$A$ variants, and our *LoLoRA* method with HPCA updates. Results are presented in Table 4, which are averaged over three runs with different seed values. We report best validation perplexity, loss, and peak extra GPU memory as the difference between the peak allocated memory and the memory required to store the frozen base model parameters in bfloat16. Additionally, we report average run time.

Table 4: Fine-tuning performance on LLaVA-v1.5-7B (Visual Instruct 150K (20% subset), one epoch).

| Method | Perplexity | Loss | Extra Memory (GB) | Run Time |
|---|---|---|---|---|
| LoRA (unifrom) | 2.90 ±0.01 | 1.066 ±0.004 | 24.6 | **2h 45m** |
| LoRA (EVA) | **2.89 ±0.01** | **1.062 ±0.003** | 24.6 | 3h 24m |
| LoRA-FA (unifrom) | 2.97 ±0.01 | 1.087 ±0.003 | **23.9** | 2h 46m |
| LoRA-FA (EVA) | 2.92 ±0.01 | 1.070 ±0.004 | **23.9** | 3h 24m |
| *LoLoRA HPCA* | 2.93 ±0.01 | 1.075 ±0.002 | 24.1 | 2h 52m |
| *LoLoRA HPCA* (EVA) | 2.93 ±0.01 | 1.074 ±0.004 | 24.1 | 3h 30m |

**Summary.** All LoRA variations achieve similar performance, but all are better than a simple freezing with random initialization. *LoLoRA HPCA* achieves lower loss compared to LoRA-FA, but HPCA updates do not improve EVA-initialized adapters. The run time highlights significant overhead for EVA initialization, while our method reaches a compromise between efficiency and performance. In this task, memory gains are not that prominent due to the short textual part, generated by the decoder, in comparison to image tokens processed by the visual encoder.

## 5.4 ABLATIONS: INITIALIZATION AND LOCAL RULES (TINYLLAMA-1.1B, ALPACA)

In this subsection we present the results of ablations for different initializations on LoRA-FA and local rules for *LoLoRA*.

**Initializations (LoRA-FA).** We compare Kaiming uniform, orthogonal, PiSSA (Meng et al., 2024) and EVA (Explained Variance Adaptation) from (Paischer et al., 2024). In all ranks, EVA gives the best final validation perplexity compared to other initializations. The results of the experiments are presented in the Table 5.

Table 5: LoRA-FA: comparison of $A$ initializations (TinyLlama-1.1B, Alpaca). The best result in each rank is shown in bold.

| Init | $r = 2$ | $r = 4$ | $r = 8$ |
|------|---------|---------|---------|
| LoRA-FA (Uniform) | $2.566\pm0.010$ | $2.554\pm0.011$ | $2.543\pm0.011$ |
| LoRA-FA (Orthogonal) | $2.567\pm0.012$ | $2.554\pm0.011$ | $2.543\pm0.011$ |
| LoRA-FA (PiSSA) | $2.572\pm0.012$ | $2.558\pm0.012$ | $2.547\pm0.012$ |
| LoRA-FA (EVA) | $\mathbf{2.558\pm0.011}$ | $\mathbf{2.546\pm0.011}$ | $\mathbf{2.536\pm0.010}$ |

**Local rules (*LoLoRA*).** We consider the following options for updating $A$: (i) HPCA - stream SNL with running mean subtraction (smoothing factor $0.98$); (ii) HPCA no mean - same without centering; (iii) HPCA (svd first) - on the first batch we perform PCA decomposition of input vectors and initialize $A$, followed by SNL steps. (iv) AE - local gradient steps from the autoencoder loss with input centering. (v) SoftHebb - a soft variation of the Hebbian rule from (Moraitis et al., 2022). For reference, we also present standard LoRA fine-tuning results (Full LoRA). On Alpaca, the most stable and best results are obtained by HPCA (svd first), classic HPCA and AE, HPCA no mean is a bit inferior, SoftHebb shows the worst performance. The results of the experiments are presented in the Table 6.

Table 6: Variants of local update rules $A$ (*LoLoRA*, TinyLlama-1.1B, Alpaca). The best result in each rank is shown in bold.

| Method | $r = 2$ | $r = 4$ | $r = 8$ |
|--------|---------|---------|---------|
| HPCA (svd first) | $\mathbf{2.557\pm0.011}$ | $2.546\pm0.011$ | $\mathbf{2.535\pm0.011}$ |
| HPCA (uniform) | $\mathbf{2.557\pm0.011}$ | $\mathbf{2.545\pm0.011}$ | $\mathbf{2.535\pm0.011}$ |
| HPCA no mean | $2.561\pm 0.011$ | $2.550\pm0.011$ | $2.540\pm0.011$ |
| AE (uniform) | $2.558\pm0.011$ | $2.547\pm0.011$ | $2.536\pm0.011$ |
| SoftHebb (uniform) | $2.573\pm0.011$ | $2.574\pm0.011$ | $2.572\pm0.011$ |
| Full LoRA (uniform) | $2.537\pm0.013$ | $2.528\pm0.013$ | $2.521\pm0.012$ |

Overall, all local update rules that converge to the optimal PCA subspace of the inputs perform equally well. Similarly, LoRA-FA with EVA initialization achieves comparable performance. However, online methods have the advantage of not requiring a separate incremental PCA pass before training.

## 6 CONCLUSION

In this work, we present a new theoretically grounded method for LLM fine-tuning (*LoLoRA*), which aims to increase fine-tuning efficiency by employing localized updates, which do not require back-propagation. We theoretically prove that the optimal initialization for the $A$ matrix in LoRA should approximate maximum eigenspace transformation. This is in line with an effective data-driven initialization EVA Paischer et al. (2024), while we propose an iterative algorithm *LoLoRA* that leads to a similar subspace during training. Our experiments showed that HPCA consistently outperforms standard LoRA-FA in two out of three experimental setups.

One of the limitations of our theoretical analysis is that we considered each submodule isolated with stationary targets, which is not strictly the case in multilayer architecture. The future work could be directed towards addressing non-stationarity. Also our method introduces a small amount of extra optimizer state for the local updates, unlike standard LoRA-FA. Another promising direction of future work is beyond standard fine-tuning: anywhere the model projects from high to low dimension (e.g., projection blocks in efficient attention variants like MLA (Ji et al., 2025)), the same local-update trick can avoid storing input activations and further cut memory.

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

## A PROOFS

### A.1 PROOF OF THEOREM 4.4

Let us write out the derivative of $B$:

$$\frac{\partial L}{\partial B} = \mathbb{E}\left(\tau - BAz\right) z^\top A^\top = \left(\Sigma_{\tau z} - BA\Sigma_{zz}\right) A^\top. \tag{3}$$

Using the necessary optimality condition for equation 3 we get

$$BA\Sigma_{zz}A^\top = \Sigma_{\tau z}A^\top. \tag{4}$$

Suppose that

$$A\Sigma_{zz}^{1/2} = U\Lambda V^\top, \tag{5}$$

where $U^\top U = V^\top V = I_d$ and $\Lambda$ are non-singular diagonal matrices. Note that

$$\Sigma_{\tau z} = \mathbb{E}\Delta W_0 zz^\top = \Delta W_0 \Sigma_{zz}. \tag{6}$$

Hence, using equation 4, equation 5 and equation 6 we obtain:

$$BU\Lambda V^\top V\Lambda U^\top = \Delta W_0 \Sigma_{zz}^{1/2} V\Lambda U^\top.$$

Simplifying the expression and multiplying both parts on the right side by $U\Lambda^{-1}V^\top$ we get

$$BU\Lambda V^\top = \Delta W_0 \Sigma_{zz}^{1/2} VV^\top.$$

Using equation 4 we derive

$$BA\Sigma_{zz}^{1/2} = \Delta W_0 \Sigma_{zz}^{1/2} VV^\top. \tag{7}$$

Let us rewrite the loss in a more convenient form

$$L = \frac{1}{2}\mathbb{E}||\tau - \Delta W_0 z||^2 = \frac{1}{2}\mathbb{E}||\tau||^2 - \mathbb{E}\langle\tau, \Delta W_0 z\rangle + \frac{1}{2}\mathbb{E}\langle\Delta W_0 z, \Delta W_0 z\rangle$$

$$= \frac{1}{2}\mathbb{E}||\tau||^2 - \langle\Sigma_{\tau z}, \Delta W_0\rangle + \frac{1}{2}\langle\Delta W_0\Sigma_{zz}, \Delta W_0\rangle$$

$$= \frac{1}{2}\mathbb{E}||\tau||^2 - \frac{1}{2}\langle\Sigma_{\tau z}, \Delta W_0\rangle - \frac{1}{2}\langle\Sigma_{\tau z} - \Delta W_0\Sigma_{zz}, \Delta W_0\rangle$$

$$= \frac{1}{2}\mathbb{E}||\tau||^2 - \frac{1}{2}\langle\Sigma_{\tau z}, \Delta W_0\rangle - \frac{1}{2}\left\langle \underbrace{\left(\Sigma_{\tau z} - BA\Sigma_{zz}\right)A^\top}_{=0}, B\right\rangle$$

$$= \frac{1}{2}\mathbb{E}||\tau||^2 - \frac{1}{2}\langle\Sigma_{\tau z}, \Delta W_0\rangle.$$

Denote $R(A) = R = \langle\Sigma_{\tau z}, \Delta W_0\rangle$. Using equation 6 and equation 7, rewrite $R$ as follows

$$R = \langle\Delta W_0\Sigma_{zz}, BA\rangle = \left\langle\Delta W_0\Sigma_{zz}^{1/2}, BA\Sigma_{zz}^{1/2}\right\rangle = \left\langle\Delta W_0\Sigma_{zz}^{1/2}, \Delta W_0\Sigma_{zz}^{1/2}VV^\top\right\rangle$$

$$= \left\langle\Sigma_{zz}^{1/2}\Delta W_0^\top \Delta W_0\Sigma_{zz}^{1/2}, VV^\top\right\rangle.$$

Hence, taking the expectation with respect to $W_0$ we obtain:

$$\mathbb{E}_{\Delta W_0}R = \mathbb{E}_{\Delta W_0}\left\langle\Sigma_{zz}^{1/2}\Delta W_0^\top \Delta W_0\Sigma_{zz}^{1/2}, VV^\top\right\rangle = \sigma^2 n\left\langle\Sigma_{zz}, VV^\top\right\rangle.$$

Therefore, by Wielandt minimax formula, which provides an extreme characterization of the sum of the largest eigenvalues of a Hermitian matrix in terms of its inner products with rank-one projections, we obtain

$$\max_A \mathbb{E}_{\Delta W_0}R(A) = \sigma^2 n \sum_{k=1}^{r} \lambda_k. \tag{8}$$

The columns of matrix $V$ define the subspace on which the maximum of the expression $\text{tr}_V(\Sigma_{zz}) = \text{tr}(V^\top\Sigma_{zz}V)$ is reached. Also, it is straightforward to see that

$$\mathbb{E}_{\Delta W_0,z}||\tau||^2 = \sigma^2 n \sum_{k=1}^{n} \lambda_k. \tag{9}$$

Therefore, using equation 8 and equation 9 we obtain 1. Using supposition equation 5 and the fact that $V$ reaches the maximum of $\text{tr}(V^\top\Sigma_{zz}V)$ on the set $\{V \mid V^\top V = I_r\}$, we get 2. If $\Sigma_{zz}$ has different eigenvalues, then $\text{range}(V) = \text{range}(u_1, u_2, \ldots, u_r)$ where $u_k$ are the eigenvectors of $\lambda_k$, hence we get 3.

## A.2  PROOF OF THEOREM 4.5

Let us write out the derivative of $A$:

$$\frac{\partial L}{\partial A} = \mathbb{E}B^\top \left(\tau - BAz\right) z^\top = B^\top \left(\Sigma_{\tau z} - BA\Sigma_{zz}\right). \tag{10}$$

Using the necessary optimality condition for equation 10 we get

$$B^\top BA\Sigma_{zz} = B^\top \Sigma_{\tau z}. \tag{11}$$

Suppose that

$$B = U\Lambda V^\top, \tag{12}$$

where $U^\top U = V^\top V = I_d$ and $\Lambda$ are non-singular diagonal matrices. Note that

$$\Sigma_{\tau z} = \mathbb{E}\Delta W_0 z z^\top = \Delta W_0 \Sigma_{zz}. \tag{13}$$

Hence, using equation 11, equation 12 and equation 13 we obtain:

$$V\Lambda U^\top U\Lambda V^\top A\Sigma_{zz} = V\Lambda U^\top \Delta W_0 \Sigma_{zz}.$$

Simplifying the expression and multiplying both parts on the right side by $U\Lambda^{-1}V^\top$ we get

$$BA\Sigma_{zz} = UU^\top \Delta W_0 \Sigma_{zz}. \tag{14}$$

Let us rewrite the loss in a more convenient form

$$L = \frac{1}{2}\mathbb{E}||\tau - \Delta W_0 z||^2 = \frac{1}{2}\mathbb{E}||\tau||^2 - \mathbb{E}\left\langle \tau, \Delta W_0 z\right\rangle + \frac{1}{2}\mathbb{E}\left\langle \Delta W_0 z, \Delta W_0 z\right\rangle$$

$$= \frac{1}{2}\mathbb{E}||\tau||^2 - \frac{1}{2}\left\langle \Sigma_{\tau z}, \Delta W_0\right\rangle - \frac{1}{2}\left\langle \underbrace{B^\top \left(\Sigma_{\tau z} - BA\Sigma_{zz}\right)}_{=0}, A\right\rangle$$

$$= \frac{1}{2}\mathbb{E}||\tau||^2 - \frac{1}{2}\left\langle \Sigma_{\tau z}, \Delta W_0\right\rangle.$$

Denote $R(B) = R = \left\langle \Sigma_{\tau z}, \Delta W_0\right\rangle$. Using equation 13 and equation 14, rewrite $R$ as follows

$$R = \left\langle \Delta W_0 \Sigma_{zz}, BA\right\rangle = \left\langle \Delta W_0, BA\Sigma_{zz}\right\rangle = \left\langle \Delta W_0, UU^\top \Delta W_0 \Sigma_{zz}\right\rangle$$

$$= \left\langle \Delta W_0 \Sigma_{zz} \Delta W_0^\top, UU^\top\right\rangle.$$

Hence, taking the expectation with respect to $\Delta W_0$ we obtain:

$$\mathbb{E}_{\Delta W_0} R = \mathbb{E}_{\Delta W_0}\left\langle \Delta W_0 \Sigma_{zz} \Delta W_0^\top, UU^\top\right\rangle = \sigma^2 \operatorname{tr}(\Sigma_{zz})\left\langle I_n, VV^\top\right\rangle = \sigma^2 \operatorname{rk}(U)\operatorname{tr}(\Sigma_{zz})$$

$$= \sigma^2 \operatorname{rk}(B)\operatorname{tr}(\Sigma_{zz}).$$

Using equation 9 we get 1. Also, it is straightforward to get 2.

## A.3  PROOF OF THEOREM 4.6

Let us rewrite our functional as follows:

$$l(A) = \mathbb{E}_z||z - A^\top Az||^2 = \mathbb{E}\left\langle z - A^\top Az, z - A^\top Az\right\rangle$$

$$= \left\langle I - A^\top A, (I - A^\top A)\Sigma_{zz}\right\rangle = ||(I - A^\top A)\Sigma_{zz}^{1/2}||^2.$$

Now let us consider what eigenvalues the matrix $A^\top A$ can have, assuming that $A$ is a local minimum of $l$. We express $A$ using the singular value decomposition: $A = UHV^\top$, where $U \in \mathbb{R}^{r \times r}$ with $U^\top U = I_r$, $H = \operatorname{diag}(h_1, \ldots, h_r)$, and $V \in \mathbb{R}^{n \times r}$ with $V^\top V = I_r$. Then $A^\top A = VH^2V^\top$. Let $V = [v_1, \ldots, v_r]$. Let us rewrite $l(A)$:

$$||(I_n - A^\top A)\Sigma_{zz}^{1/2}||^2 = ||(I_n - VH^2V^\top)\Sigma_{zz}^{1/2}||^2$$

$$= ||V(I_r - H^2)V^\top \Sigma_{zz}^{1/2} + (I_n - VV^\top)\Sigma_{zz}^{1/2}||^2$$

$$= ||V(I_r - H^2)V^\top \Sigma_{zz}^{1/2}||^2 + ||(I_n - VV^\top)\Sigma_{zz}^{1/2}||^2$$

$$+ 2\langle V(I_r - H^2)V^\top \Sigma_{zz}^{1/2}, (I_n - VV^\top)\Sigma_{zz}^{1/2}\rangle$$

$$= ||V(I_r - H^2)V^\top \Sigma_{zz}^{1/2}||^2 + ||(I_n - VV^\top)\Sigma_{zz}^{1/2}||^2$$

$$+ 2\langle (I_n - VV^\top)V(I_r - H^2)V^\top, \Sigma_{zz}\rangle$$

$$= ||V(I_r - H^2)V^\top \Sigma_{zz}^{1/2}||^2 + ||(I_n - VV^\top)\Sigma_{zz}^{1/2}||^2.$$

Consider the expression $||V(I_r - H^2)V^\top \Sigma_{zz}^{1/2}||^2$:

$$||V(I_r - H^2)V^\top \Sigma_{zz}^{1/2}||^2 = ||(I_r - H^2)V^\top \Sigma_{zz}^{1/2}||^2$$
$$= \text{tr}((I_r - H^2)V^\top \Sigma_{zz} V(I_r - H^2))$$
$$= \sum_{i=1}^{r}(1 - h_i^2)^2 v_i^\top \Sigma_{zz} v_i.$$

It follows that $h_i^2 = 1$ for any $i$ ($v_i^\top \Sigma_{zz} v_i \neq 0$ due to non-singularity of $\Sigma_{zz}$). Therefore, $AA^\top = I_r$. Now, let us rewrite $l(A)$:

$$l(A) = ||(I_n - A^\top A)\Sigma_{zz}^{1/2}||^2$$
$$= ||\Sigma_{zz}||^2 - 2\left\langle \Sigma_{zz}^{1/2}, A^\top A\Sigma_{zz}^{1/2}\right\rangle + \left\langle A^\top A\Sigma_{zz}^{1/2}, A^\top A\Sigma_{zz}^{1/2}\right\rangle$$
$$= ||\Sigma_{zz}||^2 - \left\langle \Sigma_{zz}^{1/2}, A^\top A\Sigma_{zz}^{1/2}\right\rangle$$
$$= ||\Sigma_{zz}||^2 - \text{tr}(A^\top \Sigma_{zz} A).$$

Therefore, $A^\top A = I_r$. Let $A = \begin{bmatrix} a_1 & \dots & a_r \end{bmatrix}^\top$. If $A \notin \mathcal{V}$, then by the min-max theorem, there exists a vector $v$ such that $||v|| = 1$, $\langle v, a_i \rangle = 0$, and $v^\top \Sigma_{zz} v > a_i^\top \Sigma_{zz} a_i$ for all $i$. Hence, consider the function $p(t) = l(A(t))$, where

$$A(t) = \begin{bmatrix} \cos(t)a_1 + \sin(t)v & a_2 & \dots & a_r \end{bmatrix}^\top.$$

Then $p'(0) < 0$, which contradicts the assumption that $A$ is a local minimum. Therefore, $A \in \mathcal{V}$. It is straightforward to verify that such an $A$ is a global minimizer.

## B    CONVERGENCE ANALYSIS

### B.1    LEARNING CURVES

In this section, we present learning curves for LLaVA experiments from Section 5.3 (see Figure 2). To quantify HPCA convergence, we also measure the local submodule's input recall error, which is $\mathbb{E}_z \frac{1}{2}||z - A^\top Az||^2$. It can be seen that HPCA updates stabilize quickly, and in the case of EVA initialization as a starting point, recall error increases. This is due to the non-stationarity of submodules' input caused by model training. To show that, we conducted additional experiments with frozen $B = 0$, i.e., when layer outputs are unaffected by training and remain stationary. As can be seen from Figure 3, in that case there is no such increase in recall error, and HPCA updates converge to the same recall error as for EVA initialization. Nevertheless, our further analysis shows that online HPCA updates can remedy this non-stationarity problem slightly (see Appendix B.2 for details).

### B.2    OPTIMAL SUBSPACE DISTANCE

In this section we illustrate why *LoLoRA* updates may be more useful in comparison to freezing $A$ with EVA initialization. For this reason we use chordal distance (Ye & Lim, 2016) to the optimal subspace as a metric to measure to what extent a subspace formed by matrix $A$ is far from the optimal subspace derived from SVD decomposition of a dataset for a downstream task. Chordal distance is one of the possible ways to quantify the difference between multidimensional subspaces. It is easy to compute in comparison to other similar metrics, and in the case of orthonormal matrices, it is reduced to $d(A, A') = ||AA^T - A'(A')^T||_F$, where $|| \cdot ||_F$ is the Frobenius norm.

Previous experiments are clear that in most cases, initial optimal subspace initialisation is sufficient for good performance. However, it can be shown that due to the multilayer structure of the transformer model, the input distribution of each submodule is non-stationary. The result is that the optimal subspace changes while adapters are training, as can be seen from Figure 4. It shows that the chordal distance between the initial optimal subspace and the current optimal subspace increases with training. It can be seen that the distance increases most rapidly at the very beginning of training, when the loss decreases the most.

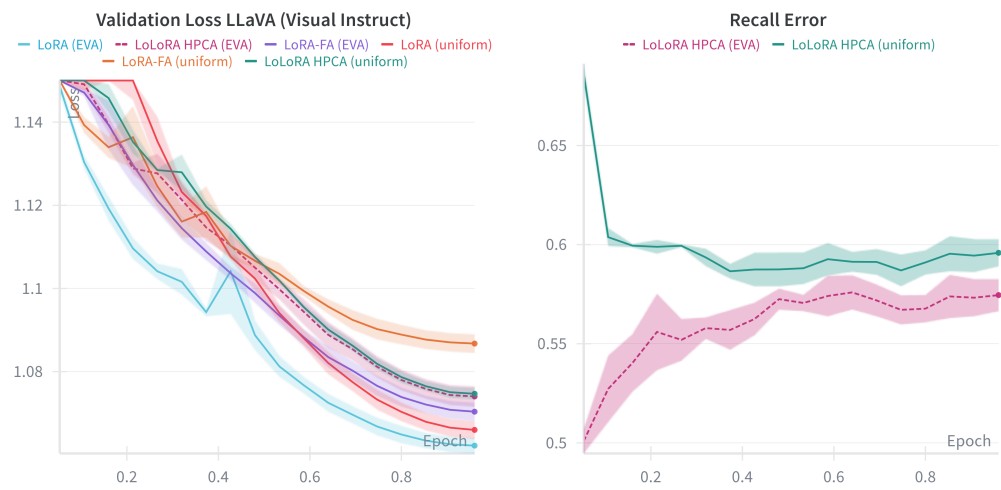

Figure 2: Learning cures for LLaVA-v1.5-7B fine-tuning during one epoch on 20% subset of Visual Instruct 150K dataset. Colored regions show one standard deviation.

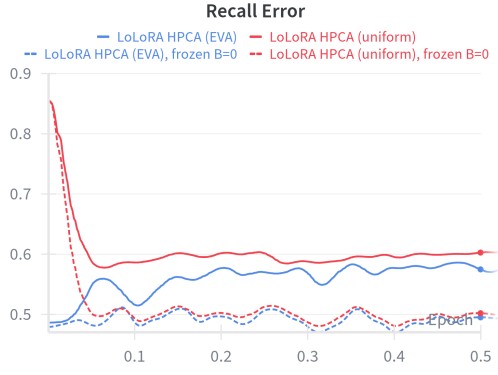

Figure 3: HPCA convergence test. Recall error for matrix $A$ in LLaVA-v1.5-7B fine-tuning experiment compared to frozen $B = 0$ case (i.e. without finetuning, only $A$ updates).

Our method allows us to partially mitigate this problem and adjust the matrix $A$ locally to bring it closer to the optimal subspace at each timestep. To show that, we fine-tune TinyLLaMA on the Alpaca dataset with the LoRA $A$ matrix initialised with the optimal subspace. In each of several steps, we calculate the SVD of the input module's activations and compare the chordal distance between the optimal subspace derived from the SVD, which is the same as the EVA initialisation, and the current $A$ matrix's subspace. As can be seen from Figure 5, the initial optimal EVA initialisation subspace quickly becomes distant from the current optimal subspace, which can be seen from the curve for the EVA-FA (LoRA-FA with EVA initialization) method. At the same time, *LoLoRA* methods (AE and HPCA) significantly reduce the distance, mitigating the effect of non-stationarity of inputs due to $B$ updates. It also may be seen that standard LoRA training also yields a subspace that is more distant from the optimal one than for other methods. The results are evident that *LoLoRA* drives the LoRA $A$ matrix's subspace to the optimal one.

## C    DETAILS OF EXPERIMENTAL SETUPS AND HYPERPARAMETERS

In this section, we detail the most important parameters for our experiments presented in Section 5.

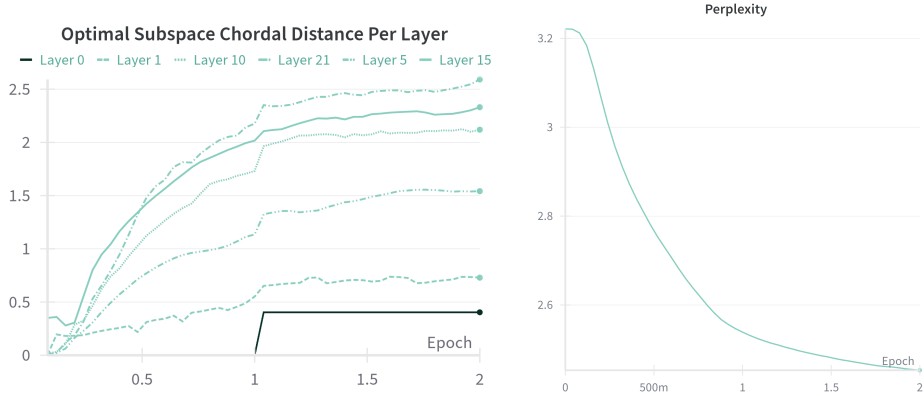

Figure 4: The distance between the initial optimal subspace and the current one increases for each layer with training (TinyLLaMA LoRA-FA (EVA) fine-tuning on Alpaca dataset). The distance is measured for $A$ matrix subspace in LoRA adapter of the key projection $W_k$. Other submodules have similar trend.

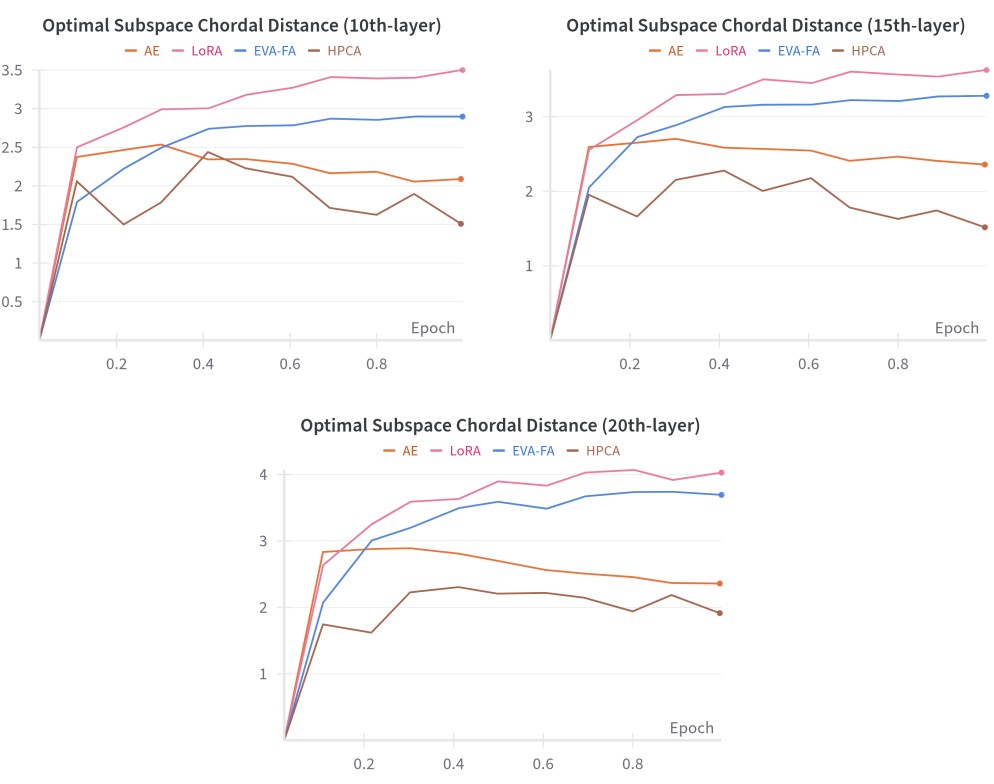

Figure 5: Chordal distance of the $W_k$ matrix's $A$ adapter from the optimal subspace increases with training for different layers. Initially all the baselines are initialised with EVA, which gives the optimal subspace. However, due to $B$ training, optimal subspace changes, and the distance increases. Our methods allow for softening this effect.

Two most important hyperparameters for *LoLoRA HPCA* updates are learning rate (for forward pass updates of $A$) and whether we should use one-batch SVD initialization to speed-up initial convergence (see Table 7). In all experiments, we scale HPCA updates similarly to Adam with

$\beta_1 = 0.9$ and $\beta_2 = 0.999$ and input centering with running average (smoothing factor 0.98), which doesn't influence forward pass.

General hyperparameters as LoRA rank ($r$) and scale ($\alpha$), batch size, maximum sequence length and number of fine-tuning epochs are presented in Table 8.

For each experiment, we conduct a grid search to define the optimal AdamW learning rate for updates on backward pass (see Table 9). Same for all scenarios: AdamW ($\beta_1 = 0.9$, $\beta_2 = 0.999$), dropout=0.0, weight_decay=0.0 with 80 steps learning rate warmup. For MetaMathQA (Section 5.2) and Visual Instruct (Section 5.3) we also employ cosine scheduler.

Table 7: HPCA parameters.

| Experiment | Learning Rate | One-Batch SVD Initialization |
|---|---|---|
| GLUE (Section 5.1) | 1.0e-4 | Yes |
| MetaMathQA (Section 5.2) | 2.5e-4 | No |
| Visual Instruct (Section 5.3) | 1.5e-3 | No |

Table 8: General parameters.

| Experiment | $r$ | $\alpha$ | Batch Size | Seq. Length | N Epoch |
|---|---|---|---|---|---|
| GLUE (Section 5.1) | 8 | 16 | 64, 16 (CoLA, RTE, MRPC, STS-B) | 512 | 5, 15 (QQP) |
| MetaMathQA (Section 5.2) | 8 | 16 | 32 | 1024 | 1 |
| Visual Instruct (Section 5.3) | 32 | 32 | 16 | 2048 | 1 |

Table 9: Optimal learning rates for AdamW optimizer (backward pass) derived using grid search.

| Experiment | LoRA (uniform \| EVA) | LoRA-FA (uniform \| EVA) | *LoLoRA HPCA* |
|---|---|---|---|
| MetaMathQA | 1.1e-5 | 3.3e-5 \| 1.7e-5 | 1.7e-5 |
| Visual Instruct | 8.0e-4 \| 3.9e-4 | 8.0e-4 \| 3.9e-4 | 3.9e-4 |
| CoLA | 3.0e-4 | 6.0e-4 \| 6.0e-5 | 3.0e-5 |
| RTE | 1.0e-4 | 6.0e-4 \| 6.0e-5 | 3.0e-5 |
| MRPC | 3.0e-4 | 3.0e-4 \| 6.0e-5 | 6.0e-5 |
| STS-B | 1.0e-4 | 6.0e-4 \| 3.0e-5 | 3.0e-5 |
| MNLI | 1.0e-4 | 1.0e-4 \| 3.0e-5 | 3.0e-5 |
| QNLI | 1.0e-4 | 3.0e-4 \| 3.0e-5 | 3.0e-5 |
| QQP | 1.0e-4 | 6.0e-4 \| 3.0e-5 | 3.0e-5 |
| SST-2 | 3.0e-4 | 6.0e-4 \| 6.0e-5 | 3.0e-5 |

**Ablations.** In all ablations we train 7 epochs, batch size = 6, sequence length = 2048, sequence packing enabled. Use constant LR with 80 warmup steps, max_grad_norm = 1.0, scaling_factor = 1 (alpha = rank). We fine-tune all attention linear layers $W_q, W_k, W_v, W_o$. For each experiment, we first do a grid-search by LR on one seed, then fix the best LR and run it on four more seeds. The procedure is repeated for ranks $r \in \{2, 4, 8\}$. All local optimizers are AdamW with standard hyperparameters ($\beta_1 = 0.9$, $\beta_2 = 0.999$). For all methods except HPCA (svd first), we set $lr_A = 1.5 \times 10^{-3}$; HPCA (svd first) uses $lr_A = 1.0 \times 10^{-4}$.

# D MEMORY MEASUREMENTS ON ROBERTA-LARGE

We conducted additional experiments to measure the difference in memory consumption on RoBERTa-large in various experimental setups (LoRA, LoRA-FA, and *LoLoRA*) on different datasets. We measured peak memory allocation using

`torch.cuda.max_memory_allocated()`. Peak memory usage during training typically comprises model parameters, adapter weights, optimizer states, activations, gradients, and temporary calculation buffers. In all setups, the model occupies 1.32GB of memory. In all setups, additional LoRA parameters ($A$ and $B$ adapters) occupy 10MB. In standard LoRA and *LoLoRA* setups, optimizer states take up 20MB, while in LoRA-FA setup, optimizer states take up 14MB. Table 10 reports peak memory usage, excluding static model weights, LoRA parameters, and optimizer states.

Table 10: Peak memory overhead (activations, gradients, and temporary buffers) in GB on RoBERTa-large across GLUE tasks. These values exclude static model weights, LoRA parameters, and optimizer states.

| Method | CoLA | RTE | MRPC | STS-B | MNLI | QNLI | QQP | SST-2 |
|---|---|---|---|---|---|---|---|---|
| LoRA | 1.23 | 4.52 | 1.96 | 2.22 | 23.48 | 27.79 | 17.40 | 4.38 |
| LoRA-FA | 1.11 | 3.64 | 1.65 | 1.85 | 18.40 | 21.79 | 13.65 | 3.48 |
| *LoLoRA HPCA* | 1.11 | 3.65 | 1.65 | 1.85 | 18.41 | 21.79 | 13.65 | 3.49 |

**The Use of Large Language Models (LLMs)** In this study, large language models were employed to polish and refine the texts.

