# OpenReview forum: "LoLoRA: Locally Fine-Tuned Low Rank Adapters"
_ICLR.cc/2026/Conference — Submitted to ICLR 2026_

### Official Review · Reviewer_aPNU · 2025-10-26

**Soundness:** 3
**Presentation:** 3
**Contribution:** 2
**Rating:** 4
**Confidence:** 4

**Summary:**

This paper proposes LoLoRA (Locally Fine-Tuned Low-Rank Adapters), a modification of LoRA that introduces local unsupervised updates for the adapter matrix A to reduce activation memory usage during fine-tuning. Instead of keeping Acompletely frozen as in LoRA-FA, LoLoRA applies Hebbian PCA (HPCA) or autoencoder-based local updates in the forward pass while updating B through backpropagation. The authors provide a theoretical analysis showing that optimal A initialization aligns with the principal subspace of the input covariance matrix and validate this through experiments on RoBERTa-Large (GLUE), LLaVA-v1.5-7B, and TinyLlama-1.1B.

**Strengths:**

1.	Clear theoretical grounding: The paper provides a well-formalized theoretical motivation for the proposed local learning rule, which rigorously characterize optimal conditions for the adapter matrices.
2.	Readable structure and comprehensive related work: The manuscript is well-organized, situating LoLoRA among recent LoRA variants (EVA, LoRA-FA, Adalora, etc.) and giving clear algorithmic descriptions.

**Weaknesses:**

1.	The main motivation is to improve LoRA-FA by locally updating the frozen A matrix, but the improvement appears incremental and not conceptually substantial. The paper may be seen as a minor extension of LoRA-FA with limited novelty.
2.	The background explanation of why freezing A reduces GPU memory is brief and could be clarified with a more detailed analysis or quantitative breakdown of memory components saved (activations vs. optimizer states).
3.	Experimental results, especially in Table 3, show negligible GPU memory savings (24.6 GB → 24.1 GB). It is unclear whether the trade-off justifies the added algorithmic complexity.
4.	The models and training scale are relatively small. For the multimodal experiment, only one epoch is trained on 10% of LLaVA 150K. This leaves uncertainty about convergence and whether LoLoRA would maintain its advantage under full-scale training or with more epochs.

**Questions:**

1.	What do the convergence curves look like for different methods (LoRA, LoRA-FA, LoLoRA)? Is there a noticeable difference in early-stage convergence speed?
2.	Since freezing or locally updating A yields limited memory savings, can the authors quantify the exact memory components saved (e.g., activation cache, gradient tensors, optimizer states)?
3.	In LoLoRA, the local optimizer introduces extra states for A; how significant is this overhead compared to the saved activation memory?

---

> ### Author Response · Authors · 2025-11-26
>
> Thank you for your comments and constructive feedback and your appreciation of the theoretical grounding of our work.
>
> ## [W1] Limited novelty.
> We appreciate this remark, it helps us to improve the presentation of our work.
> We should emphasize that the contribution of our work lies in advancing existing approaches, such as PCA-based adapter initialization and local/unsupervised learning. While these components exist individually, their integration into a unified framework has not been undertaken before and represents a natural evolution in this area of research. Furthermore, we believe that the idea of local updates is conceptually significant, and the question of “which modules can be trained locally and according to what rules” deserves more attention. The observation that local signals can yield competitive performance, even without relying on global backpropagation, is noteworthy.
>
> ## [W2, 3; Q 2, 3] Memory breakdown and clarifying memory savings.
> Thank you for these important questions.
> Peak memory usage during training typically comprises model parameters, adapter weights, optimizer states, activations, gradients, and temporary calculation buffers. Regarding memory consumption, LoRA and LoLoRA HPCA are practically equivalent in terms of optimizer states (local HPCA states are comparable in size to global optimizer moments for matrix $A$). The main savings in LoLoRA HPCA are achieved by excluding matrix $A$ from the backpropagation graph. Specifically, there is no need to cache the layer input data to calculate the gradient on $A$. In addition, the tensor $\nabla A$ is not created or stored in memory during the backward pass (the update occurs on the forward pass).
>
> | Method | Activation Memory Savings | Gradient Memory Savings | Optimizer State Savings |
> | :--- | :---: | :---: | :---: |
> | **LoRA** | No | No | No |
> | **LoRA-FA** | Yes | Yes | Yes |
> | **LoLoRA (Ours)** | Yes  | Yes  | No |
>
> The memory costs for the optimizer states and gradients for matrix $A$ are O(hidden\_size * lora\_rank). At the same time, the memory required to store input activations scales as O(batch * hidden\_size * sequence\_length). Typically, batch * sequence\_length >> lora\_rank, so the main benefit is achieved by avoiding activation caching.
>
> We also have added additional memory usage measurements that more clearly demonstrate the benefits of our approach. They can be found in **Points 2 and 3** of the General Response. Please, see also **Section 5.2** for additional experiments that show up to 13% memory savings for LLaMA-3 and the memory measurements in **Appendix D** that demonstrate up to 20% memory savings for RoBERTa-Large.
>
> ## [W4] The models and training scale are relatively small.
> We conducted additional experiments on LLaVA-v1.5-7b, demonstrating performance trends on the extended training dataset. Also, we conducted new experiments on fine-tuning LLaMA-3.1-8B-Instruct and validated on GSM8K Platinum. The results of the experiments can be found in **Points 3 and 4** of the General Response and **Sections 5.2, 5.3** of the revised manuscript.
>
> ## [Q1] Convergence curves.
> Please refer to **Appendix B.1, Figure 2** of the updated version, where we provide the convergence plot of the algorithm. According to the results, we think that there is not much difference in the convergence speed; however LoRA-FA shows slightly faster convergence to a suboptimal result. We should note that learning rates (see **Table 9 in Appendix C**) are tuned for each method using grid search, which might compensate for these differences.

---

### Official Review · Reviewer_fDLi · 2025-10-31

**Soundness:** 2
**Presentation:** 1
**Contribution:** 2
**Rating:** 2
**Confidence:** 4

**Summary:**

The paper introduces LoLoRA, a modification of LoRA designed to reduce activation memory usage during fine-tuning. It updates the adapter matrix $A$ locally (via a Hebbian-style or PCA-like rule, or AE) during the forward pass, avoiding backpropagation and storage of input activations. Adapter $B$ is updated normally through backprop. The authors provide an analysis showing that the initialization is important and conduct experiments on GLUE, LLaVA, and TinyLLaMA to validate the method's performance and efficiency.

**Strengths:**

- The paper connects LoRA initialization with PCA-based optimal subspaces under a random regression model, offering insight into the asymmetry between $A$ and $B$ adapters.
- Introducing local updates for LoRA aligns with current efforts to reduce memory in PEFT.
- The problem this paper tackles is essential.

**Weaknesses:**

- The empirical results are not good enough. The claimed performance-efficiency tradeoff is not convincing, as LoLoRA's memory savings are minor (on the order of a few hundred MB) and do not justify the added algorithmic complexity; also, it underperforms.
- The experiments are limited to very low ranks ($r={2,4,8}$).
- The theory gives intuition but lacks empirical validation under realistic conditions.
- The paper requires major writing improvements; phrasing needs to be improved, and organization is uneven.
- Comparison to more baselines is needed; e.g., to other memory-efficient PEFT methods (e.g., orthonormal LoRA, rank selection, and so on). The current baseline selection is limited.
- No runtime or compute cost analysis is given.

**Questions:**

Please refer to the above weaknesses.

---

> ### Author Response · Authors · 2025-11-26
>
> Thank you for recognizing the theoretical insights of our work and your constructive feedback.
>
> ## [W1, 3] The empirical results are not good enough.
> We have added a discussion on this matter in Point 1 of the General Response. We have provided additional measurements in Point 2 of the General Response that rigorously demonstrate the memory savings of our approach. Please, see also our newer results in Point 3 of the General Response and Section 5.2 of the revised manuscript.
>
> ## [W2] The experiments are limited to very low ranks (r=2, 4 ,8).
> We respectfully point out that our LLaVA experiments were conducted with LoRA rank $r=32$, demonstrating memory gains and competitive performance. We conducted additional experiments, emphasizing performance trends on an extended training dataset. The results of the experiments can be found in Point 4 of the General Response and Section 5.3 of the revised manuscript.
> ## [W4] Writing improvements
> We substantially revised text in that regard, improving phrasing and structure by removing technical details to Appendix C.
>
> ## [W5] Comparison to more baselines is needed.
> In **Table 5**, which compares different initializations for LoRA-FA on TinyLlama-1.1B (Alpaca), we added a new baseline (PiSSA [1]) in the revised version of the paper.
>
> **Table 5.** LoRA-FA: comparison of *A* initializations (TinyLlama-1.1B, Alpaca). Best in each rank is **bold**.
> | Init | r=2 | r=4 | r=8 |
> |---|---:|---:|---:|
> | LoRA-FA (Uniform) | 2.566 ± 0.010 | 2.554 ± 0.011 | 2.543 ± 0.011 |
> | LoRA-FA (Orthogonal) | 2.567 ± 0.012 | 2.554 ± 0.011 | 2.543 ± 0.011 |
> | **LoRA-FA (PiSSA)** | 2.572 ± 0.012 | 2.558 ± 0.012 | 2.547 ± 0.012 |
> | LoRA-FA (EVA) | **2.558 ± 0.011** | **2.546 ± 0.011** | **2.536 ± 0.010** |
>
> ## [W6] No runtime or compute cost analysis is given.
> Please see our discussion on compute/runtime complexity in Point 5 of the General Response. Additionally, we report run time for LLaVA experiments in **Section 5.3** of the revised paper and Point 4 of the General Response.
>
> ***
> [1] Fanxu Meng, Zhaohui Wang, and Muhan Zhang. Pissa: Principal singular values and singular vectors adaptation of large language models. arXiv preprint arXiv:2404.02948, 2024.

---

### Official Review · Reviewer_DFhv · 2025-10-31

**Soundness:** 3
**Presentation:** 2
**Contribution:** 3
**Rating:** 4
**Confidence:** 3

**Summary:**

LoRA was developed to reduce the memory requirements for a model, reducing the size of the trained parameters at fine-tuning. However, the size of the activations is the same for fine-tuning as for LoRA. LoLoRA solves this by not storing the inputs to each linear module $z$ but instead freezes parameter $A$ and stores $Az$. Building on the work of LoRA-FA, A is initialized in a more theoretically-grounded way to assist in training. This matrix then is updated locally to regain the expressiveness of LoRA training both $A$ and $B$. Theory is developed to determine what the best update to $A$ would be, relating it to the eigen-decomposition of the empirical covariance matrix. This method is then tested against standard LoRA and LoRA where A is frozen on language models and language-vision models.

**Strengths:**

- The core concept of keeping A frozen to reduce memory requirements significantly while also updating A through non-gradient methods is very appealing. Prior work keeping A frozen is clearly improved upon by learning A without the same scale of gradient as is needed for standard LoRA
- The updates to A are theoretically grounded, leading to some nice theoretical understanding of what A will be learned
- All experiments are run multiple times, where every experiment has its error bars clearly listed

**Weaknesses:**

- Parts of the writing are quite hard to follow. The abstract mentions the "adapter" A which wasn't clear in its meaning as one of the two LoRA matrices. There are also references to "local updates", which seem to mean updates that do not involve a gradient; this understanding only came after reading much of the paper in depth
- The method is named LoLoRA, yet this name is unused in the tables, making it difficult to parse what are the results to be compared against. Using the name LoLoRA-HPCA would be much clearer
- The results aren't very strong, with Table 1 and 2 both showing that these method is typically of worse performance than simply freezing A
- Only after a number of reads was it noticed that there are no figures. While not necessary, a figure depicting how LoLoRA works and its back-and-forth between optimizing B with gradients and optimizing A with this local update would be invaluable

**Questions:**

- Can the memory requirements for Table 1 and 2 also be shown? In those situations, having a comparison of required memory would lend stronger credibility to LoLoRA as it uses less memory than LoRA
- What is being shown in Table 4 and 5? They seem like perplexities but it’s unclear

---

> ### Author Response · Authors · 2025-11-26
>
> Thank you for your insightful comments and remarks, and appreciation of the rigor of our work.
>
> ## [W1] Clarification on terminology
> Thank you for this remark. We revised the manuscript and added clarifications about the terminology used. Based on your comment, we also revised the abstract to make the scope of our work more clear.
>
> ## [W2] Method names in tables
> Thank you for noticing that. We adjusted method names in tables and text to make them more consistent.
>
> ## [W3] "The results aren't very strong"
> We have added a discussion on this matter in point 1 of the General Response. Please, see also our additional results in the General Response.
>
> ## [W4] Adding a diagram
> Thank you for this suggestion. We added a diagram in the revised version of the manuscript.
>
> ## [Q1] Can the memory requirements for Table 1 and 2 also be shown?
> Please, find them in Point 2 of the General Response. We also added **Appendix D** in the revised manuscript to address this question.
>
> ## [Q2] What is being shown in Table 4 and 5?
> Yes, perplexity. We will add clarification in the final version.

---

### Official Review · Reviewer_cEpi · 2025-11-01

**Soundness:** 4
**Presentation:** 3
**Contribution:** 3
**Rating:** 8
**Confidence:** 3

**Summary:**

This paper proposes a new LoRA finetuning method which reduces the memory usage. The idea is to use a local update rule for training the A matrix. Prior work has suggested freezing matrix but this leads to reduced performance. The method in this paper aims to bridge that gap. With the local update rule, the activations generated by A no longer need to be saved. The local update is performed using a Hebbian learning rule which in an online fashion estimates the top-r eigenspace of the input activation covariance. Using the top-r eigenspace has been previously suggested as a data-drive initialization for the A matrix. The authors prove the optimality of this setting of the A matrix under an isotropic prior for the ideal update. The B matrix is trained normally. The authors perform a variety of experiments demonstrating that their algorithm can preserve the performance of vanilla LoRA while reducing the memory usage.

**Strengths:**

The idea is simple and clearly motivated. The experiments are thorough. The performance does not seem to degrade when using the local rule.

**Weaknesses:**

The isotropic assumption on $\Delta W$ is a bit of a stretch. That said, the resulting update still seems reasonable and it unclear if there are more meaningful and tractable assumptions.

In a small setting, there could be a sanity check that the top-r eigenspace is being learned accurately by the online algorithm and there can be a comparison when the true exact eigenspace is used.

Some of the theoretical arguments could be less dense and written more clearly.

**Questions:**

How does the step time compare with vanilla LoRA?

---

> ### Author Response · Authors · 2025-11-26
>
> Thank you for your appreciation of our contribution and thoughtful questions. We updated our manuscript to address your questions and concerns.
>
> # About isotropic assumption on $\Delta W$.
> Thank you for your question. It concerns how to model the ignorance of the optimal change to the weight matrices of a pretrained model during fine-tuning. On the one hand, before fine-tuning, we already have some information relevant to the new task, such as the pretrained weights $W$ themselves. On the other hand, it's not clear how to correctly use this information to specify a prior distribution on $\Delta W$. One natural option is to assume that $\Delta W$ is structurally related to $W$. Practical work by PiSSA [1], OLoRA [2], and DoRA [3] has already addressed this issue. Notably, the original LoRA paper [4] did not find a strong correlation between the leading eigenvectors of the matrices $\Delta W$ and $W$ (see Section H.3 “CORRELATION BETWEEN W AND ∆W”). Therefore, we use the isotropic assumption of $\Delta W$ as a “null hypothesis” in situations where reliable structural information about the prior is unavailable. Also, It is significant that this assumption yields a class of methods already known in practice (EVA-like initialization). We consider the direction of refining the assumptions of $\Delta W$ to be an interesting field for further theoretical research.
>
> # Sanity check of HPCA convergence.
> Thank you for your question. The HPCA convergence plots show an interesting effect: during fine-tuning, the input distribution to the layers becomes more anisotropic. **Figure 3 in Appendix B** of the updated manuscript shows the reconstruction error convergence for four settings, LoLoRA HPCA, LoLoRA HPCA with EVA initialization and the same with frozen $B=0$. We observe that fine-tuning changes the layer-input distributions, increasing their anisotropy. In this setting of frozen $B=0$, the LoRA adapters do not affect the activations, so the input distribution remains unchanged, which allows us to fairly compare how well each method captures the fixed input distribution.
>
> We also highlight that we have added a more detailed analysis of the algorithm's convergence and comparison to the ground true eigenspace in **Appendix B.2** of the revised paper.
>
> # How does the step time compare with vanilla LoRA?
> This is indeed an important question. We have added a discussion on compute/runtime complexity in Point 5 of the General Response. In short, we argue that LoLoRA is on par with LoRA in that regard, however, further engineering optimizations are possible to improve the efficiency of our method.
>
> **References**
>
> [1] Fanxu Meng, Zhaohui Wang, and Muhan Zhang. Pissa: Principal singular values and singular vectors adaptation of large language models. arXiv preprint arXiv:2404.02948, 2024.
> [2] Kerim Büyükakyüz. OLoRA: Orthonormal Low-Rank Adaptation of Large Language Models. arXiv preprint arXiv:2406.01775, 2024.
> [3] Yulong Mao, Kaiyu Huang, Changhao Guan, Ganglin Bao, Fengran Mo, and Jinan Xu. DoRA: Enhancing Parameter-Efficient Fine-Tuning with Dynamic Rank Distribution. arXiv preprint arXiv:2405.17357, 2024.
> [4] Edward J. Hu, Yelong Shen, Phillip Wallis, Zeyuan Allen-Zhu, Yuanzhi Li, Shean Wang, Lu Wang, and Weizhu Chen. LoRA: Low-Rank Adaptation of Large Language Models. In International Conference on Learning Representations (ICLR), 2022.

---

### Author Response · Authors · 2025-11-26
**General Response Part 1**

We thank all reviewers for their constructive feedback and the time invested in reviewing our paper. We are encouraged by the recognition of our method’s sound theoretical grounding (Reviewers **cEpi**, **DFhv**, **aPNU**) and the clarity of the motivation (**cEpi**, **aPNU**).

We revised our paper and uploaded the updated version of the manuscript that includes changes addressing your concerns and questions. Below, we address the common concerns raised by reviewers.

# 1. Сoncerns regarding Empirical Results (Reviewers **DFhv**, **fDLi**)
Regarding the comment that the results are “not very strong”, we emphasize that LoLoRA HPCA consistently achieves parity with the LoRA-FA (EVA) (which uses data-driven SVD initialization) and outperforms standard random initialization (LoRA-FA) on generative tasks. Crucially, EVA initialization is an offline method. It requires a separate preliminary pass through the data to calculate eigenvectors (preprocessing). Our method achieves the same advantages online by adapting to the input distribution directly during training. We present the results of our extended experiments below.

# 2. Additional memory measurements on RoBERTa-Large  (Reviewers **DFhv**, **fDLi**, **aPNU**)
We conducted additional experiments to measure the difference in memory consumption on Roberta-large in various experimental setups (LoRA, LoRA-FA, and LoLoRA HPCA) on different datasets, which are reflected in **Appendix D of the revised manuscript**. We measured peak memory allocation using `torch.cuda.max_memory_allocated()`. Peak memory usage during training typically comprises: model parameters, adapter weights, optimizer states, activations, gradients, and temporary calculation buffers. In all setups, the model occupies 1.32GB of memory. In all setups, additional LoRA parameters ($A$ and $B$ adapters) occupy 10MB. In classic LoRA and LoLoRA HPCA setups, optimizer states take up 20MB, while in LoRA-FA setup, optimizer states take up 14MB. The table below reports peak memory usage, excluding static model weights, LoRA parameters, and optimizer states:

**Table 1: Peak memory overhead (activations, gradients, and temporary buffers) in GB on RoBERTa-large across GLUE tasks.**
*These values exclude static model weights, LoRA parameters, and optimizer states.*

| Method | CoLA | MNLI | MRPC | QNLI | QQP | RTE | SST-2 | STS-B |
| :--- | :---: | :---: | :---: | :---: | :---: | :---: | :---: | :---: |
| **LoRA** | 1.23 | 23.48 | 1.96 | 27.79 | 17.40 | 4.52 | 4.38 | 2.22 |
| **LoRA-FA** | 1.11 | 18.40 | 1.65 | 21.79 | 13.65 | 3.64 | 3.48 | 1.85 |
| **LoLoRA HPCA** | 1.11 | 18.41 | 1.65 | 21.79 | 13.65 | 3.65 | 3.49 | 1.85 |

In half of the experiments, batch_size=64, in the other half batch_size=16, in all experiments max_seq_length=512 is set, which limits the maximum number of tokens. However, most samples do not reach this limit due to their short length, which is further compressed by the tokenizer.

**Conclusion:** The LoRA-HPCA and LoRA-FA setups in the Roberta-large experiments have approximately the same peak memory values, which are significantly lower than the values in the LoRA setup, with the difference increasing as max_seq_length * batch_size increases.

# 3. LLaMA-3.1-8B additional experiments (Reviewer **aPNU**)
We conducted additional experiments on LLaMA-3.1-8B-Instruct fine-tuned on MetaMathQA and evaluated on the GSM8K Platinum benchmark. **Please see Table 3** in the revised text, we also provide it here for convenience:

| Method | Accuracy | Extra Memory (GB) |
| :--- | :---: | :---: |
| **LoRA (uniform)** | 0.821 $\pm$ 0.005 | 30 |
| **LoRA-FA (uniform)** | 0.826 $\pm$ 0.005 | 26 |
| **LoRA-FA (EVA)** | 0.829 $\pm$ 0.005 | 26 |
| **LoLoRA HPCA (uniform)** | 0.829 $\pm$ 0.004 | 26 |
| **Base** | 0.79 | - |

Training was conducted for a single epoch, LoRA rank is 8, alpha=16. Maximum sequence length was set to 1024 and batch size was 32. As shown, LoRA-FA, LoRA-FA (EVA), and LoLoRA HPCA achieve approximately 13\% reduction in VRAM (26 GB vs 30 GB, excluding model weights) compared to standard LoRA while maintaining or improving performance. Both LoRA-FA (EVA) and LoLoRA HPCA achieve the highest accuracy of 82.9\%, representing a 3.9 percentage point improvement over the base model.

---

### Author Response · Authors · 2025-11-26
**General Response Part 2**

# 4. LLaVA-v1.5-7B extended experiments (Reviewers **fDLi**, **aPNU**)
We conducted extended experiments on fine-tuning LLaVA-v1.5-7B on 20% of the LLaVA Visual Instruct 150K dataset. The updated results are presented in **Table 4 of the revised manuscript.**

**Table: LLaVA-v1.5-7B Fine-Tuning on 20% (30k) of LLaVA Visual Instruct 150K dataset, validation on 1.5k subset**
| Method        |   Perplexity   |       Loss       | Extra Memory, GB |  Time  |
|---------------|:--------------:|:----------------:|:--------:|:------:|
| LoRA (uniform)          | 2.90 $\pm$0.01 | 1.066 $\pm$0.004 |   24.6   | 2h 45m |
| LoRA (EVA)      | 2.89 $\pm$0.01 | 1.062 $\pm$0.003 |   24.6   | 3h 24m |
| LoRA-FA (uniform)       | 2.97 $\pm$0.01 | 1.087 $\pm$0.003 |   23.9   | 2h 46m |
| LoRA-FA (EVA)   | 2.92 $\pm$0.01 | 1.070 $\pm$0.004 |   23.9   | 3h 24m |
| LoLoRA HPCA (uniform)     | 2.93 $\pm$0.01 | 1.075 $\pm$0.002 |   24.1   | 2h 52m |
| LoLoRA HPCA (EVA) | 2.93 $\pm$0.01 |  1.074 $\pm$0.00 |   24.1   | 3h 30m |

As we can see, the trend of LoLoRA HPCA, LoRA-FA (EVA), and LoRA (EVA) advantages persist. In the updated version, we also present average run time for each method, which highlights significant overhead for EVA initialization, while our method reaches a compromise between efficiency and performance. We also should note that in this task, memory gains are not that prominent due to the **short textual part**, generated by the decoder, in comparison to image tokens processed by the visual encoder.

# 5. Runtime / compute cost analysis (Reviewers **cEpi**, **fDLi**)
From a theoretical perspective, compared to LoRA-FA, LoLoRA HPCA performs an additional large matrix multiplication $y^{\top}x$ on the forward pass. LoRA, in contrast, performs the multiplication  $\partial y^{\top}x$ to calculate the global gradient on the backward pass. Therefore, in terms of FLOPS, these methods are comparable. Although the local update can be computed in parallel with the forward pass, our current implementation performs it sequentially for simplicity. We believe that further engineering optimizations could bridge the remaining gap in wall-clock time. However, the primary focus of this work is on algorithmic validation.

---

### Meta-Review · Area_Chair_UhRi · 2026-01-06

**Summary:**

I recommend a reject due to: i) A few reviewers complained about the poor writing and presentation of the paper. After checking the revised manuscript, I think this problem persists. ii) Reviewers mentioned that the empirical results do not show much improvements over baselines. I agree with that. iii) Reviewers mention that the benefit of the method in terms of memory savings is not obvious. While the authors have addressed this, the results seem to not have been derived under comprehensive experiments. In all, the paper seems like a rushed submission. The only strong positive rating (8) was overly kind.

**Reviewer Concerns:**

The reviewers mentioned about the poor writing and presentation, as well as lacklustre empirical results. I do not think these concerns were well-addressed.

**Reviewer Scores:**

The score should remain the same. I do not think the recommended rejections can be overturned.

---

### Decision · Program_Chairs · 2026-01-26

Reject